# Use of Carbon Nanoparticles to Improve Soil Fertility, Crop Growth and Nutrient Uptake by Corn (*Zea mays* L.)

**DOI:** 10.3390/nano11102717

**Published:** 2021-10-14

**Authors:** Fengliang Zhao, Xiaoping Xin, Yune Cao, Dan Su, Puhui Ji, Zhiqiang Zhu, Zhenli He

**Affiliations:** 1Indian River Research and Education Center, Department of Soil and Water Science, Institute of Food and Agricultural Sciences, University of Florida, Fort Pierce, FL 34945, USA; zfl7409@163.com (F.Z.); xinxp1024@ufl.edu (X.X.); caohua3221@163.com (Y.C.); sudan1980@126.com (D.S.); jipuhui1983@163.com (P.J.); zhuzhiq8@163.com (Z.Z.); 2Hainan Key Laboratory of Tropical Eco-Circular Agriculture, Institute of Environmental and Plant Protection, Chinese Academy of Tropical Agricultural Sciences, Haikou 571101, China

**Keywords:** nanomaterial, crop growth, nutrient availability, enzyme activity, sandy soil

## Abstract

The use of carbon nanoparticles (CNPs) as a fertilizer synergist to enhance crop growth has attracted increasing interest. However, current understanding about plant growth and soil response to CNPs is limited. In the present study, we investigated the effects of CNPs at different application rates on soil properties, the plant growth and nutrient use efficiency (NUE) of corn (*Zea mays* L.) in two agricultural soils (Spodosol and Alfisol). The results showed that CNPs affected corn growth in a dose-dependent manner, augmenting and retarding growth at low and at high concentrations, respectively. The amendment at the optimal rate of 200 mg CNPs kg^−1^ significantly enhanced corn growth as indicated by improved plant height, biomass yield, nutrient uptake and nutrient use efficiency, which could be explained by the higher availability of phosphorus and nitrogen in the amended soils. The application of CNPs largely stimulated soil urease activity irrespectively of soil types. However, the responses of dehydrogenase and phosphatase to CNPs were dose dependent; their activity significantly increased with the increasing application rates of CNPs up to 200 mg kg^−1^ but declined at higher rates (>400 mg kg^−1^). These findings have important implications in the field application of CNPs for enhancing nutrient use efficiency and crop production in tropical/subtropical regions.

## 1. Introduction

The increase in global population, combined with improved income and dietary changes, is driving an ever-increasing food demand that is expected to rise by 70% between now and 2050 [1]. The application of traditional fertilizers has been instrumental for increasing agricultural productivity, but efficacy has significantly dropped since the ‘green revolution’ and the excessive application of chemical fertilizers has resulted in negative effects on soil health, leading to low nutrient utilization efficiency and environmental pollution [2]. In addition, climate change, land degradation and urbanization have posed more challenges to sustainable agriculture and a search for novel materials and technologies has become more urgent in modern agriculture [3].

Nanotechnology has shown great potential in terms of enhancing plant growth due to small size, large surface area and the high reactivity of nanoparticles (NPs) [4]. Intensive studies have focused on nano-fertilizers (NFs) and/or nano-additives that could serve as nutrient or fertilizer synergist to enhance seed germination rates, plant growth, crop yield and soil quality [5,6]. Carbon nanoparticles (CNPs) were applied to augment crop production [7], seed germination [8], cell division [9,10], water uptake [11] and photosynthesis [12]. For instance, the number of tomato fruits and flowers was doubled when the plants were treated with carbon nanotubes (CNTs) [7]. Tomato seed priming with graphene increased the content of chlorophyll, vitamin C, β-carotene, phenols, flavonoids, and H_2_O_2_ content by 111%, 78%, 11 folds, 85%, 45% and 215% in seedlings, respectively [8].

Carbon NPs are composed of pure carbon and therefore exhibit high stability, low toxicity and environmental friendliness [13]. Our current understanding of plant growth and soil responses to CNPs is limited [14]. Investigations into the applications and fate of CNPs in crop and soil remain scarce and are mostly limited to laboratory scales [4]. Most agricultural soils in Florida are extremely sandy with sand content often in excess of 90% [15] and are characteristic of the low content of nutrient-retaining soil constituents (clay, organic matter and the oxides of Fe and Al) and readily subjected to leaching loss [16]. In the present study, it was hypothesized that CNPs may have potential in promoting plant growth and improving soil quality, especially nutrient use efficiency (NUE) in the tropical/subtropical regions where most soils inherit low organic carbon content. The objectives of this study were to determine the effects of CNPs on: (1) corn growth and nutrient uptake; (2) soil physicochemical and biochemical properties; and (3) the optimal application rate of CNPs. The results are expected to improve our understanding of CNPs’ application potential in crop production and soil quality improvement.

## 2. Materials and Methods

### 2.1. CNPs and Corn Seeds

The carbon nanoparticles used in this study were supplied by Vulpes Agricultural Corp. (Saint Louis, MO, USA). The tested CNPs were composed of 62% C, 1.5% H, 35% O, 1.5% N, as determined by a VARIO EL III CHON analyzer (Elementar, Langenselbold, Germany). The CNPs had a size range of 20–130 nm with negative charges on surfaces (zeta potential −67.6 mV), which was determined using Malvern Nano-ZS (Malvern Instruments Ltd., Malvern, UK). The seeds of corn (*Zea mays* L., var. Early Sunglow Hybrid) were purchased from Park Seed company (Hodges, SC, USA).

### 2.2. Tested Soils

Two sandy soils under agriculture at a depth of 0–20 cm were collected from the University of Florida (UF) experimental farm (Spodosol) and McArthur citrus grove (Alfisol) located in Fort Pierce, Florida. Soil physiochemical properties were determined following the standard procedures of Soil Sampling and Methods of Analysis [17]. Spodosol soil (Ankona series) is classified as sandy, siliceous, hyperthermic, oststein Arenic Haplaquods with 91% sand, 3.6% silt and 5.5% clay; while Alfisol soil (Riviera series) is classified as loamy, siliceous, hyperthermic Arenic Glossaqualfs with 82% sand, 5.9% silt and 12% clay. Briefly, the Alfisol soil had a pH 6.22, organic matter (OM) 13.9 g kg^−1^, cation exchange capacity (CEC) 15.1 cmol_c_ kg^−1^, total N (TN) 0.42 g kg^−1^ and available P estimated by Olsen extraction method (Olsen-P) 11.2 mg kg^−1^—all of which were higher than those of Spodosol soil (pH 4.61, OM 7.76 g kg^−1^, CEC 5.68 cmol_c_ kg^−1^, TN 0.18 g kg^−1^ and Olsen-P 10.3 mg kg^−1^, respectively).

### 2.3. Greenhouse Experiments

Soils were fertilized at rates of 200 mg N kg^−1^ soil, 100 mg P kg^−1^ soil and 200 mg K kg^−1^ soil, respectively, of which 30% N, 50% P and 31.5% K were applied as base fertilizer by mixing with soil as urea, potassium dihydrogen phosphate and potassium sulfate, while the other 40% N, 50% P and 31.5% K were top-dressed to the seedlings at a height of 10 cm and the remaining N and K to the seedlings at a height of 20 cm [18].

The fertilized soils were subjected to the treatments with CNPs at the final application doses: 0, 50, 100, 200, 400 and 800 mg kg^−1^ soil (named FB0, FB50, FB100, FB200, FB400 and FB800, respectively). Soils without fertilizer or CNPs (F0B0) were used as control. Soils without fertilizer but with 200 mg kg^−1^ soil CNPs (F0B200) were used to investigate the CNPs’ effects on plant growth under non-fertilized conditions. Each treatment had five replicates.

Soil pot culture experiments were conducted in the University of Florida’s Indian River Research and Education Center greenhouse with a natural day/night light intensity, temperature of 18–26 °C and relative humidity of 60–90%. After being surface sterilized with 4% sodium hypochlorite solution, washed and soaked for 12 h, five corn seeds were sown in each pot at a depth of 2–3 cm and all the pots were arranged following a two-factor randomized block design. A week after germination, plants were thinned to one seedling per pot. All the pots were evenly watered on an “as needed” basis. Watering frequency ranged from once per week at the beginning when plants were small to three times per week towards the end of the experiment when the plants grew bigger.

### 2.4. Plant Measurements and Nutrient Analysis

To understand the dynamic effects of CNPs at different application rates, plant height was recorded on the 49th day after planting. At the end of incubation, the plants were divided into aboveground parts and roots by separately harvesting the shoot and root of corn plants from each pot. All the plant samples were rinsed with deionized water to remove adhered soil and the roots were further soaked in 10 mM EDTA for two hours and then rinsed with deionized water to remove ions or CNPs that were attached the external root surface. Plant shoots and roots were separately oven-dried at 80 °C until constant weight and dry biomass was recorded.

All the dried plant samples were ground to <0.5 mm with a stainless Wiley mill (Thomas Scientific, Swedesboro, NJ, USA). Total N in plant was determined using an 828 Series LECO automated C/N elemental analyzer (St. Joseph, MI, USA). Plant samples (0.2 g) were digested with 5 mL of concentrated nitric acid (HNO_3_) and the digested solutions were analyzed for total concentrations of P and K using inductively coupled plasma-optical emission spectrometry (ICP-OES, Horiba Instruments Inc., Edison, NJ, USA).

Nutrient use efficiency (NUE) of corn plants is calculated by the following equation:NUE = (N_treat_ − N_control_)/N_fert_,
where N_treat_ and N_control_ are the crop nutrient uptake by the aboveground biomass with and without N fertilizer, respectively, and N_fert_ is the N application rate.

### 2.5. Soil Nutrient Analysis

After being air-dried, total N concentration in soils was determined by dry combustion at 1200 °C of 50 mg samples using the LECO automated C/N elemental analyzer. Soil ammonium N (NH_4_–N) and nitrate N (NO_3_–N) were extracted with the KCl method, with the concentration of NH_4_–N and NO_3_–N in the extracts being determined with the colorimetric-indophenol blue method using a dual wavelength spectrophotometry (Hitachi U-3010, Hitachi Ltd., Irvine, CA, USA). Available P concentration was extracted using 0.5 M NaHCO_3_ and the P concentration in the extracts was determined by the ascorbic reduction-molybdenum-blue method [19].

### 2.6. Soil Enzyme Activity

Soil urease, phosphatase and dehydrogenase play very important roles in affecting the transformation and availability of soil N, P and C, respectively. Urease activity in fresh soil samples was measured by the modified method of McGarity and Myers [20]. Briefly, fresh soil (2.0 g oven-dry basis) was incubated at 30 °C for 1 h in a glass bottle after adding 1 mL toluene, 5 mL of 10% urea and 5 mL citrate buffer (pH = 6.7). After incubation, ammonium ion was extracted with 2 M KCl solution and determined by the indophenol blue method using a spectrophotometer (Hitachi U-3010, Hitachi Ltd., Irvine, CA, USA). Assays without soil were simultaneously examined as control. Soil urease activity was expressed as mg NH_4_–N kg^−1^ soil h^−1^.

Phosphatase activity in fresh soil samples was measured by the modified method according to Tabatabai et al. [21]. Briefly, fresh soils (2.0 g, oven-dry basis) were incubated at 37 °C for 1 h after adding 1 mL of toluene, 10 mL modified universal buffer (pH 6.5) and 5 mL of 100 mM p-nitrophenyl phosphate (PNP) solution. After incubation, 1 mL of 0.5 M CaCl_2_ solution and 4 mL 0.5 M NaOH solution was added. The suspensions were filtered, and the filtrate samples were immediately measured using the spectrophotometer at λ = 400 nm. Assays without soil were simultaneously examined as controls. Soil phosphatase activity was expressed as mg nitrophenyl kg^−1^ soil h^−1^.

Dehydrogenase activity in fresh soil samples was measured by the modified method of Casida [22]. Briefly, fresh soil (5.0 g oven-dry basis) was incubated in the dark at 37 °C for 24 h after adding 2 mL of 0.5% 2,3,5-triphenyl tetrazolium chloride (TTC). After incubation, 10 mL of methanol was added into the glass bottles and shaken for 1 h. The soil suspensions were filtered through a filter paper and the triphenylformazan (TPF)-formed absorbance was immediately measured at λ = 485 nm against the extracting solution using a spectrophotometer. Soil dehydrogenase activity was expressed as mg TPF kg^−1^ soil h^−1^.

### 2.7. Statistical Analysis

A two-way ANOVA (SPSS 16.0) was performed to analyze the effects of CNPs at different application rates on plant growth and the chemical and biochemical properties of soils. When the ANOVA was significant, means were compared using the Duncan multi-range test at the 95% confidence level. Differences between observations were considered statistically significant at *p* < 0.05. Figures and tables were generated with Origin 8 and Microsoft Excel 2010, respectively.

## 3. Results

### 3.1. Plant Height

Plant height increased with the increasing incubation times in both Spodosol and Alfisol soils (*p* < 0.05; Figure 1), though plants were generally higher in Alfisol than Spodosol soil, which could be associated with Alfisol soil being more fertile with higher clay, organic matter content, CEC and available nutrients. The effects of CNPs on plant height varied with application rates and soil types (*p* < 0.05). For example, the addition of 100 mg CNPs kg^−1^ in Alfisol soil significantly increased plant height to 112 cm, which was 34.9% and 14.9% higher than that in non-fertilized (F0B0) and fertilized (FB0) soils, respectively. Among all the application rates, 200 mg CNPs kg^−1^ soil (FB200) led to the highest plant height in both Spodosl and Alfisol soil, followed by FB100 and FB50 (*p* < 0.05; Figure 1). The lowest plant height was observed in FB800 for the Spodosol soil, and F0B0 for Alfisol soil (*p* < 0.05). In addition, the impact of CNPs on plant height in Spodosol soil was more intensive than that in Alfisol soil (*p* < 0.05). For example, FB200 significantly increased corn plant height by 110% in Spodosol soil than FB0; however, it was only 13.7% in Alfisol soil. The response of corn plants to CNPs was stronger in Spodosol than Alfisol soil (*p* < 0.05; Figure 1). At the end of incubation, the plant height of FB0 was significantly lower as compared to other treatments, which might result from ammonia toxicity due to urea hydrolysis. This adverse effect was significantly alleviated by the application of CNPs in the range of 50–400 mg kg^−1^ (*p* < 0.05). However, when the CNPs application was over 800 mg kg^−1^ soil, the toxicity of ammonium occurred again, probably due to the higher amount of ammonium adsorbed by CNPs (*p* < 0.05).

### 3.2. Shoot and Root Biomass

Without fertilizers, CNPs’ amendment to Spodosol soil at 200 mg kg^−1^ significantly increased root biomass (*p* < 0.05) (Figure 2). With fertilizers, both the shoot and root biomass yield of corn plants were increased with an increasing application rate of CNPs up to 200 mg kg^−1^, which then decreased at 400 mg kg^−1^ or higher for both soils (*p* < 0.05). The response of corn plant growth to CNPs amendment was generally stronger in Spodosol soil than Alfisol soil (*p* < 0.05). As compared with FB0, the combined application of 200 mg CNMs kg^−1^ soil and fertilizer led to the higher shoot biomass (313% for Spodosol soil and 25.7% for Alfisol soil) and root biomass (281% for Spodosol soil and 55.7% for Alfisol soil).

### 3.3. Nutrient Uptake and Fertilizer Use Efficiency

Although there were no significant differences in N uptake by corn plants between the treatments without fertilizers (*p* > 0.05), CNPs application significantly affected the N uptake by plants under fertilization (*p* < 0.05) (Figure 3). With fertilizers, plant N uptake dramatically increased with the increasing CNPs application rates (0–200 mg kg^−1^); however, decreased when the CNPs rate was over 400 mg kg^−1^ in both types of soils.

Without fertilizer, the CNPs‘ application did not improve P uptake by corn plants for both soils (*p* > 0.05), while the P uptake of FB200 in Spodosol soil was significantly increased compared to FB0—an increase of 267% under fertilization (*p* < 0.05). In addition, the amendment of CNPs to Alfisol soil at 800 mg kg^−1^ also increased P uptake by 24.9% compared to the control (FB0) (*p* < 0.05; Figure 4).

Without fertilizers, the addition of CNPs at 200 mg kg^−1^ significantly improved plant K uptake in Spodosol soil (*p* < 0.05)—an increase of 178%—but no significant effect occurred in Alfisol soil (*p* > 0.05) (Figure 5). When applied together with fertilizers, CNPs at adequate rates (0–200 mg kg^−1^) enhanced plant K uptake in Spodosol soil, but it was the opposite when the CNPs rate was raised to >200 mg kg^−1^ (e.g., FB400 and FB800). The largest K uptake occurred at the application rate of 200 mg CNPs kg^−1^ soil, which was 336% more than that of FB0 (*p* < 0.05). As for Alfisol soil, plant K uptake steadily increased with the increasing CNPs application rate from 200 to 800 mg kg^−1^, an increase of 19.4–28.5% compared to FB0 (*p* < 0.05; Figure 5).

Due to ammonium toxicity, the treatments of FB0 and FB800 in Spodosol soil resulted in significantly lower nutrient use efficiency (*p* < 0.05). However, the CNPs amendment to the sandy soils significantly improved fertilizer use efficiency by corn plants (*p* < 0.05; Table 1). For instance, the treatment of CNPs at 200 mg kg^−1^ in Spodosol led to the highest N, P and K use efficiency—increases of 48.8%, 11.8% and 26.7%, respectively. As for Alfisol soil, the highest use efficiency of N, P and K by corn plants occurred in FB200, FB800 and FB800, respectively—increases of 39.6%, 34.3% and 44.6%, respectively.

### 3.4. Soil Available N and P

There was no significant difference in available N and total extractable N in soil between F0B0 and F0B200 treatment (*p* > 0.05); however, the application of CNPs together with fertilizers significantly affected the total and available N in both soils during the entire growth period of corn plants (*p* < 0.05) (Figure 6). As compared with the control (FB0), soil ammonium N in Spodosol soil was significantly lower with CNPs at low application rates (50–400 mg kg^−1^), whereas ammonium N in Spodosol soil dramatically increased when the CNPs application was up to 800 mg kg^−1^. Similar results were obtained with nitrate N in soils except for a sharp rise of nitrate N occurred with CNPs application rates at 400–800 mg kg^−1^. This could be ascribed to the higher plant uptake of N at low to medium rates of CNPs and decreased plant growth at high application rates (Figure 3). Total extractable N in Spodosol soil increased with an increase in CNPs application rates; however, it significantly decreased at 800 mg kg^−1^ (*p* < 0.05).

Alfisol soil had a lower concentration of ammonium N and much higher total extractable N regardless of treatments compared with Spodosol soil (*p* < 0.05). CNPs application at 0–200 mg kg^−1^ had no effects on ammonium N in Alfisol soil (*p* > 0.05), while it was significantly increased with CNPs at the rate of 400–800 mg kg^−1^ (*p* < 0.05). As compared with FB0, nitrate N in Alfisol soil significantly increased with CNPs application at >100 mg kg^−1^; however, total extractable N significantly decreased when the CNPs application rate was up to 400–800 mg kg^−1^ (*p* < 0.05; Figure 6).

The treatments of FB0 and FB800 in Spodosol soil with fertilizer contained more available P, which were 31.5% and 59.1% higher, respectively (*p* < 0.05), as compared to FB200. Similarly, available P in Alfisol soil with FB200 was significantly lower (*p* < 0.05; Figure 7), which could be attributed to more P uptake by corn plants, as compared with FB400.

### 3.5. Soil Enzyme Activities

Soil urease activity under fertilization significantly varied with CNP application rates, though the difference was not significant between F0B0 and F0B200 without fertilizers (*p* > 0.05) (Figure 8). Urease activity in Spodosol soil sharply increased from 59.1 for the control (FB0) to 116 mg NH_4_-N kg^−1^ soil h^−1^ with CNP application at 800 mg kg^−1^—an increase of 96.7% (*p* < 0.05). In Alfisol soil, urease activity increased by 37.0% with the treatment of FB800 compared to the control (*p* < 0.05).

As compared to the treatments without fertilizers, phosphatase activity in the fertilized treatments (FB0) was significantly lower for both soils (*p* < 0.05; Figure 9). Application of CNPs at 50–200 mg kg^−1^ significantly increased phosphatase activity (by 195–239%) for Spodosol soil and 93–109% for Alfisol soil (*p* < 0.05). However, soil phosphatase activity sharply declined with CNPs at 400 mg kg^−1^ or more.

Without fertilizers, the application of CNPs at 200 mg kg^−1^ dramatically improved dehydrogenase activity—an increase of 77.0% for Spodosol soil and 44.3% for Alfisol soil compared to F0B0 (*p* < 0.05; Figure 10). Under fertilization, the application of CNPs at 100–800 mg kg^−1^ in Spodosol and 50–400 mg kg^−1^ in Alfisol soil significantly stimulated soil dehydrogenase activity (*p* < 0.05).

## 4. Discussion

### 4.1. Dose-Dependent Effect of CNP on Plant Growth

Carbon nanomaterials at low application doses were reported to activate physiological processes in plants [23]. Tiwari et al. [24] reported that carbon nanotubes (20 mg L^−1^) increased the dry weight of *Zea mays* after 7 d exposure to normal environmental conditions. Wheat (*Triticum aestivum*) plants were treated with CNPs (10–150 mg L^−1^) in the soil for up to 20 days and the results indicated that an optimum growth of wheat plants occurred at 50 mg L^−1^ treatment, with shoot and root lengths enhanced up to three folds compared to the controls [25]. In the present study, the effects of CNPs on the growth of corn plants significantly varied with application rates. Plant height, shoot and root biomass yields all markedly increased with the increasing application rate of CNPs from 0 to 200 mg kg^−1^, by 110%, 313% and 281% in Spodosol and 13.7%, 25.7% and 55.7% in Alfisol soil, respectively (Figure 1 and Figure 2). The mechanism of promoting crop growth for NPs involves the more effective uptake and transport of nutrient and water by aquaporins [26]. NPs loaded nutrient ions in the rhizosphere are transported through the following structures: epidermis, cortex and stele to reach the xylem [27]. Furthermore, NPs have a high surface area and sorption capacity and serve as reservoirs of nutrient ions with control over the rate of their release [28].

However, growth inhibition occurred at higher application rates (400–800 mg kg^−1^), especially in Spodosol soil, which might be caused by some adverse effects of the CNPs, as Spodosol soil had a lower buffering capacity than Alfisol soil because of its lower clay content. The dose-dependent effect of MWCNTs (0–1000 mg L^−1^) was observed in *Salvia verticillata* L. with a decrease in the photosynthetic pigments and increases in the oxidative stress indices (enzymatic and non-enzymatic antioxidants) in the plant leaves [29]. However, López-Vargas et al. [8] reported that seed priming with CNTs did not affect the germination rate of the tomato seeds; inversely, it negatively affected the vigor variables, such as the root length (up to 39.2%) and hypocotyl biomass (up to 33%).

### 4.2. Effects of CNPs on Plant Uptake and Use Efficiency of Nutrients

Carbon NPs play an important role in reducing the losses of nutrients and improving fertilizer use efficiency [30]. For instance, the application of CNPs on *Nicotiana tabacum* L. resulted in enhanced growth at different stages, as compared to conventional fertilizers, and increased the concentrations of N and K in plant tissues; K concentration in leaves at the maturity stage increased by 21–37% [31]. In the present study, 200 mg kg^−1^ CNPs significantly increased the plant use efficiency of N, P and K by 1891%, 609% and 597%, respectively, in Spodosol soil compared to the control, and the increased nutrient use efficiency by corn plants was probably related to the higher plant uptake, as the plant uptake of N, P and K with FB0 was very small due to ammonia toxicity that inhibited plant growth. The highest N, P and K use efficiency occurred in Alfisol soil with the treatments of FB200, FB800 and FB800, with an increase in the use of efficiency by 39.6%, 34.3% and 44.6%, respectively. The application of CNPs increased plant growth and promoted the absorption and accumulation of macro-elements in the plant, thereby increasing fertilizer use efficiency [14]. Therefore, the CNPs would be a preferable choice of manure or fertilizer alone, because of the slow and uncontrolled release of nutrients for better assimilation by the plants. Previous studies have been conducted to evaluate the optimal application rates of CNPs for different crops. However, more research is needed to understand the interactions between CNPs and plants, as there is still a knowledge gap regarding the physiological response of crop plants to CNPs, especially at relatively high application rates.

### 4.3. Effects of CNPs on Soil Available Nutrient and Enzyme Activities

Intensive use may also lead to the accumulation of CNPs in soil and consequently affect the soil physiochemical properties and microbial community, which subsequently influences the cycling of nutrients in agroecosystems [32]. In the present study, the application of CNPs at high rates improved the storage capacity of soils for available nutrients, as evidenced by increased ammonium N, nitrate N and available P in both soils, which might be related to the retention and/or complexation on the surfaces of CNPs due to their large specific surface area and carrying negative charge.

Some investigators reported that CNPs had limited impacts on the structure and functions of the soil microbial community [33,34]. However, application of CNPs at high rates (≥500 mg kg^−1^ soil) could affect microbial activity and biomass in soils [32]. As reflected by the alternations of soil enzyme activities in Figure 8, Figure 9 and Figure 10, the tested CNPs largely stimulated soil urease activity irrespectively of soil types. The application of CNPs at 50–200 mg kg^−1^ soil significantly increased phosphatase activity; however, the reverse was true at higher application rates. As compared with FB200, the phosphatase activity for the treatments with 400 mg CNPs kg^−1^ more sharply declined by 43.8–64.2% for Spodosol soil and 35.2–38.6% for Alfisol soil, respectively.

The response of dehydrogenase to CNPs was also dose dependent, especially in Alfisol soil; the dehydrogenase activity for the treatments with 400 mg CNPs kg^−1^ or more in Alfisol soil decreased by 13.9–32.4% compared with FB200. The differential response between the two soils may be associated with the differences in physical and chemical properties, including pH, organic matter and clay content, as these properties can have strong interactions with CNPs [35]. Xin et al. [11] reported that multi-walled carbon nanotubes slightly stimulated the activities of urease, phosphatase and dehydrogenase at low dosages (200 ppm) but inhibitory effects occurred at the higher dosages (500 ppm). Apparently, the effects of CNPs on soil microbial activity were dependent on their application rates and soil types. Our results suggest that high concentrations of CNPs could reduce soil enzyme activity in the tested soils, and our findings may serve as an important guideline in regulating the application of CNPs in agriculture.

## 5. Conclusions

The addition of CNPs could significantly enhance the growth of corn plants in sandy soils, as evidenced by increased shoot and root biomass yields with an optimal application rate of 200 mg kg^−1^ soil. However, the application of CNPs at higher rates (>400 mg kg^−1^) resulted in inhibitory effects on corn plant growth, likely caused by ammonium toxicity and the potential damage of CNPs on the plants. Moreover, CNPs could effectively improve the use efficiency of N, P and K by enhancing their uptake by corn plants, which may be attributed to increased nutrient retention and enzyme activities in the soils. Interestingly, CNPs had greater impacts on corn growth in Spodosol as compared to Alfisol soil, likely due to the lower soil buffering capacity of the former. Overall, the effects of CNPs on corn growth, soil nutrient availability and enzyme activities were strongly dependent on application rates and soil type.

## Figures and Tables

**Figure 1 nanomaterials-11-02717-f001:**
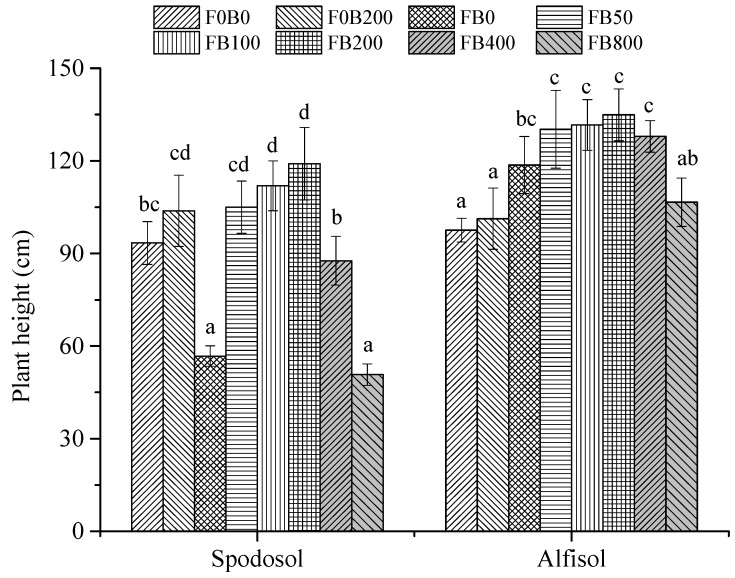
Effects of CNPs on the plant height of corn grown in Spodosol and Alfisol soils on the 49th day after transplanting. F0B0, control (without fertilizers and CNPs); F0B200, without fertilizers and with CNPs at 200 mg kg^−1^ soil; and FB0, FB50, FB100, FB200, FB400 and FB800 represent those with fertilizers and CNPs at application rates of 0, 50, 100, 200, 400 and 800 mg kg^−1^ soil, respectively. Error bars represent the standard error of the mean (*n* = 5). The letters a, b, c, and d indicate statistically significant difference at *p* < 0.05 within each group comparison between the treatments. Bars with no common letters are significantly different (*p* < 0.05).

**Figure 2 nanomaterials-11-02717-f002:**
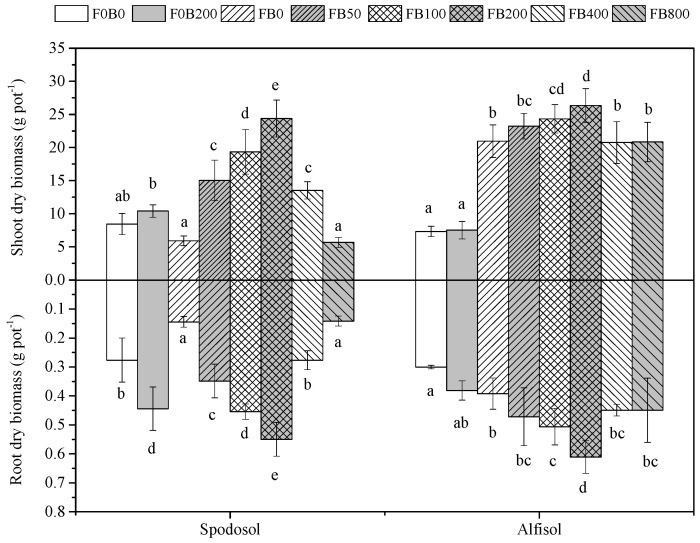
The shoot and root dry biomass of corn plants affected by different application rates of CNPs in Spodosol and Alfisol soils. F0B0, control (without fertilizers and CNPs); F0B200, without fertilizers and with CNPs at 200 mg kg^−1^ soil; and FB0, FB50, FB100, FB200, FB400 and FB800 represent those with fertilizers and CNPs at application rates of 0, 50, 100, 200, 400 and 800 mg kg^−1^ soil, respectively. Error bars represent the standard error of the mean (*n* = 5). The letters a, b, c, d and e indicate statistically significant difference at *p* < 0.05 within each group comparison between the treatments. Bars with no common letters are significantly different (*p* < 0.05).

**Figure 3 nanomaterials-11-02717-f003:**
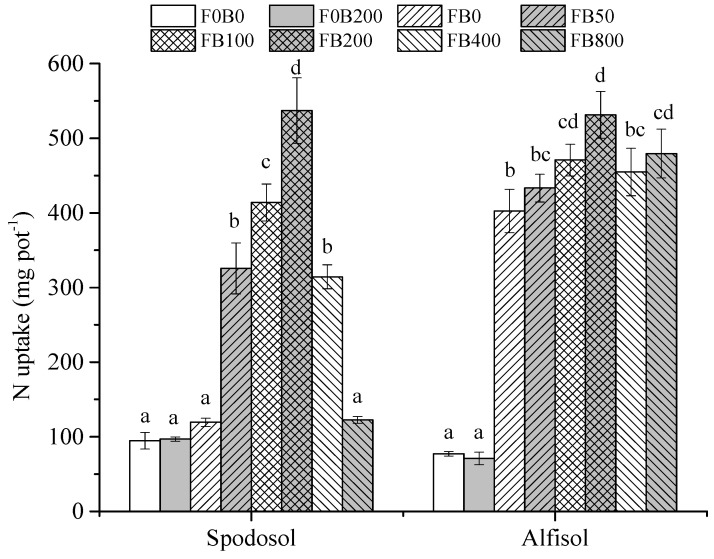
N uptake of maize plants affected by different application rates of CNPs in Spodosol and Alfisol soils. F0B0, control (without fertilizers and CNPs); F0B200, without fertilizers and with CNPs at 200 mg kg^−1^ soil; and FB0, FB50, FB100, FB200, FB400 and FB800 represent those with fertilizers and CNPs at application rates of 0, 50, 100, 200, 400 and 800 mg kg^−1^ soil, respectively. Error bars represent the standard error of the mean (*n* = 5). The letters a, b, c, and d indicate statistically significant difference at *p* < 0.05 within each group comparison between the treatments. Bars with no common letters are significantly different (*p* < 0.05).

**Figure 4 nanomaterials-11-02717-f004:**
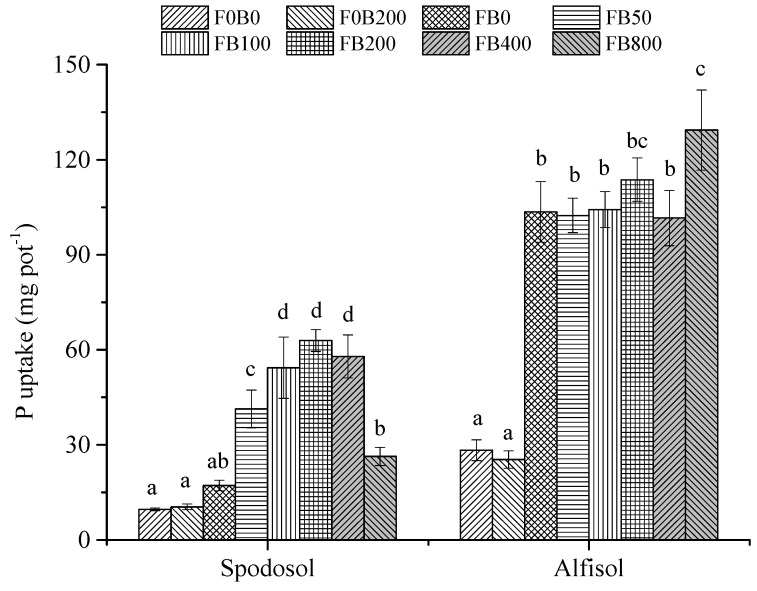
P uptake of maize plants affected by different application rates of CNPs in Spodosol and Alfisol soils. F0B0, control (without fertilizers and CNPs); F0B200, without fertilizers and with CNPs at 200 mg kg^−1^ soil; and FB0, FB50, FB100, FB200, FB400 and FB800 represent those with fertilizers and CNPs at application rates of 0, 50, 100, 200, 400 and 800 mg kg^−1^ soil, respectively. Error bars represent the standard error of the mean (*n* = 5). The letters a, b, c, and d indicate statistically significant difference at *p* < 0.05 within each group comparison between the treatments. Bars with no common letters are significantly different (*p* < 0.05).

**Figure 5 nanomaterials-11-02717-f005:**
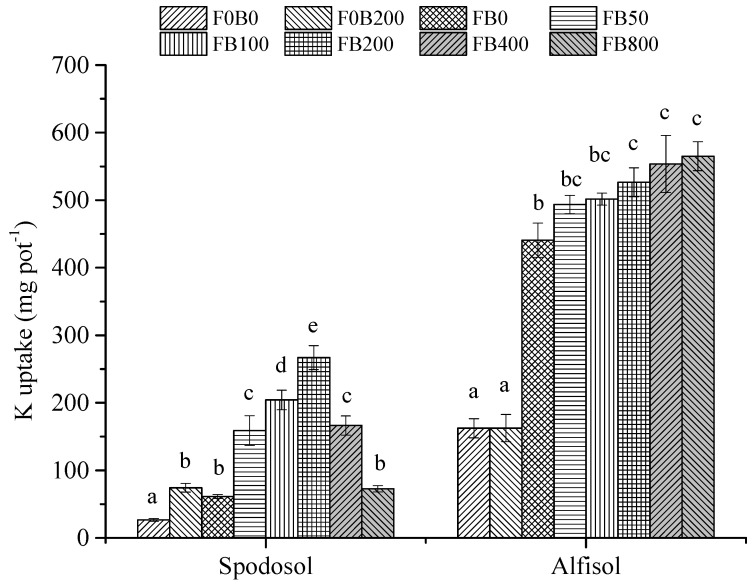
K uptake of maize plants affected by different application rates of CNPs in Spodosol and Alfisol soils. F0B0, control (without fertilizers and CNPs); F0B200, without fertilizers and with CNPs at 200 mg kg^−1^ soil; and FB0, FB50, FB100, FB200, FB400 and FB800 represent those with fertilizers and CNPs at application rates of 0, 50, 100, 200, 400 and 800 mg kg^−1^ soil, respectively. Error bars represent the standard error of the mean (*n* = 5). The letters a, b, c, d and e indicate statistically significant difference at *p* < 0.05 within each group comparison between the treatments. Bars with no common letters are significantly different (*p* < 0.05).

**Figure 6 nanomaterials-11-02717-f006:**
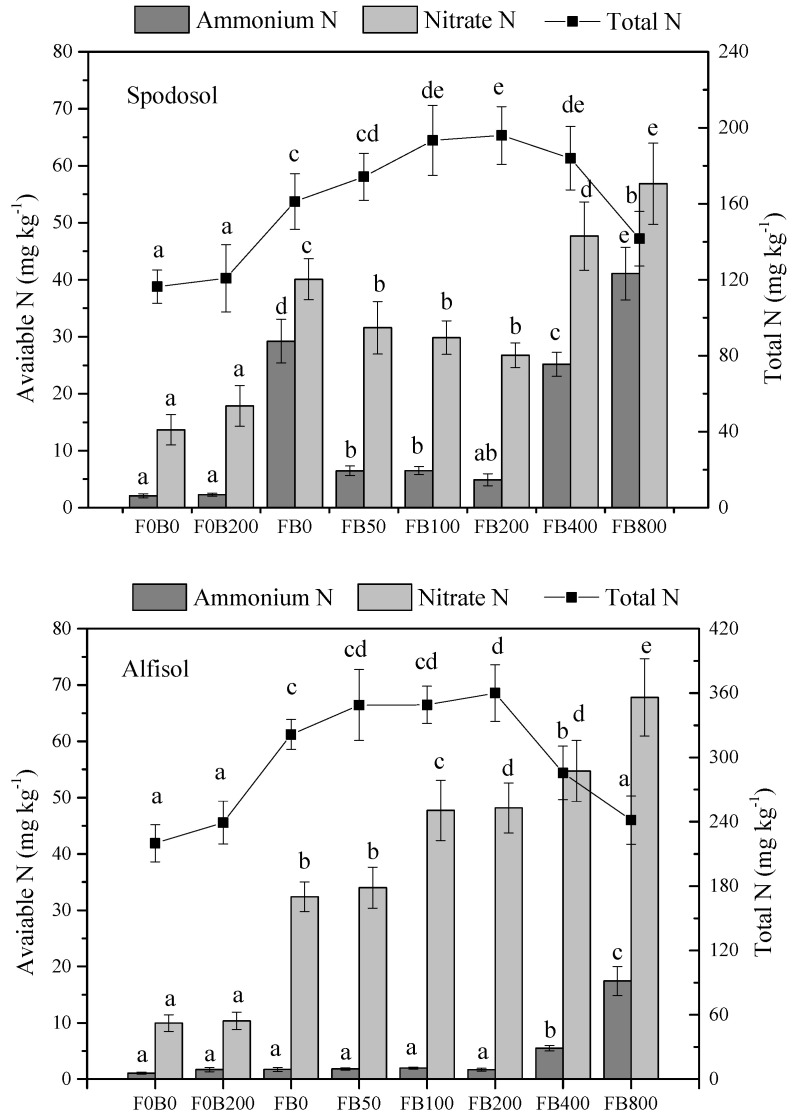
Total and available N concentration in Spodosol and Alfisol soils affected by different application rates of CNPs during the period of corn growth. F0B0, control (without fertilizers and CNPs); F0B200, without fertilizers and with CNPs at 200 mg kg^−1^ soil; and FB0, FB50, FB100, FB200, FB400 and FB800 represent those with and CNPs at application rates of 0, 50, 100, 200, 400 and 800 mg kg^−1^ soil, respectively. Error bars represent the standard error of the mean (*n* = 5). The letters a, b, c, d and e indicate statistically significant difference at *p* < 0.05 within each group comparison between the treatments. Bars with no common letters are significantly different (*p* < 0.05).

**Figure 7 nanomaterials-11-02717-f007:**
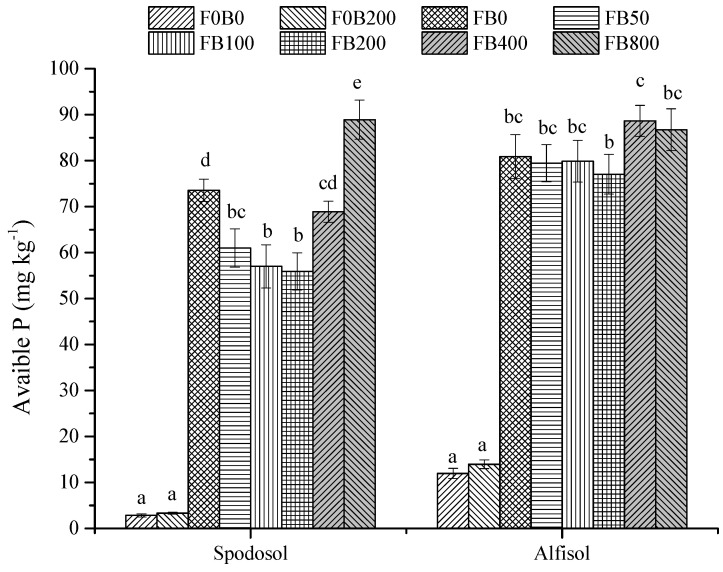
Available P in Spodosol and Alfisol soils affected by different application rates of CNPs. F0B0, control (without fertilizers and CNPs); F0B200, without fertilizers and with CNPs at 200 mg kg^−1^ soil; and FB0, FB50, FB100, FB200, FB400 and FB800 represent those with and CNPs at application rates of 0, 50, 100, 200, 400 and 800 mg kg^−1^ soil, respectively. Error bars represent the standard error of the mean (*n* = 5). The letters a, b, c, d and e indicate statistically significant difference at *p* < 0.05 within each group comparison between the treatments. Bars with no common letters are significantly different (*p* < 0.05).

**Figure 8 nanomaterials-11-02717-f008:**
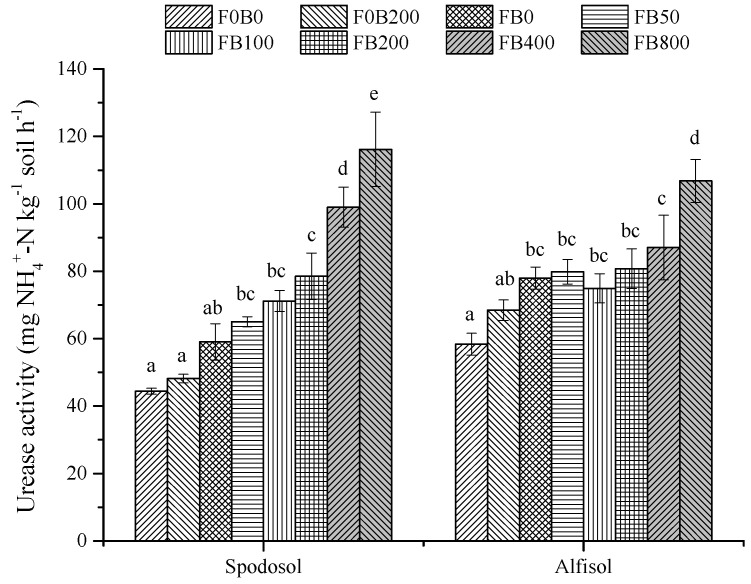
Soil urease activity (UA) in Spodosol and Alfisol soils affected by different application rates of CNPs. F0B0, control (without fertilizers and CNPs); F0B200, without fertilizers and with CNPs at 200 mg kg^−1^ soil; and FB0, FB50, FB100, FB200, FB400 and FB800 represent those with and CNPs at application rates of 0, 50, 100, 200, 400 and 800 mg kg^−1^ soil, respectively. Error bars represent the standard error of the mean (*n* = 5). The letters a, b, c, d and e indicate statistically significant difference at *p* < 0.05 within each group comparison between the treatments. Bars with no common letters are significantly different (*p* < 0.05).

**Figure 9 nanomaterials-11-02717-f009:**
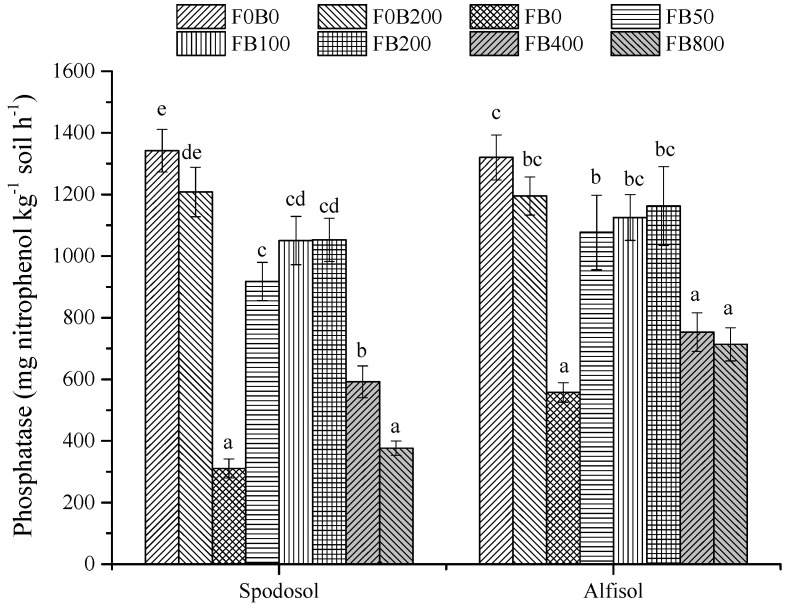
Soil phosphatase activity in Spodosol and Alfisol soils affected by different application rates of CNPs. F0B0, control (without fertilizers and CNPs); F0B200, without fertilizers and with CNPs at 200 mg kg^−1^ soil; and FB0, FB50, FB100, FB200, FB400 and FB800 represent those with and CNPs at application rates of 0, 50, 100, 200, 400 and 800 mg kg^−1^ soil, respectively. Error bars represent the standard error of the mean (*n* = 5). The letters a, b, c, d and e indicate statistically significant difference at *p* < 0.05 within each group comparison between the treatments. Bars with no common letters are significantly different (*p* < 0.05).

**Figure 10 nanomaterials-11-02717-f010:**
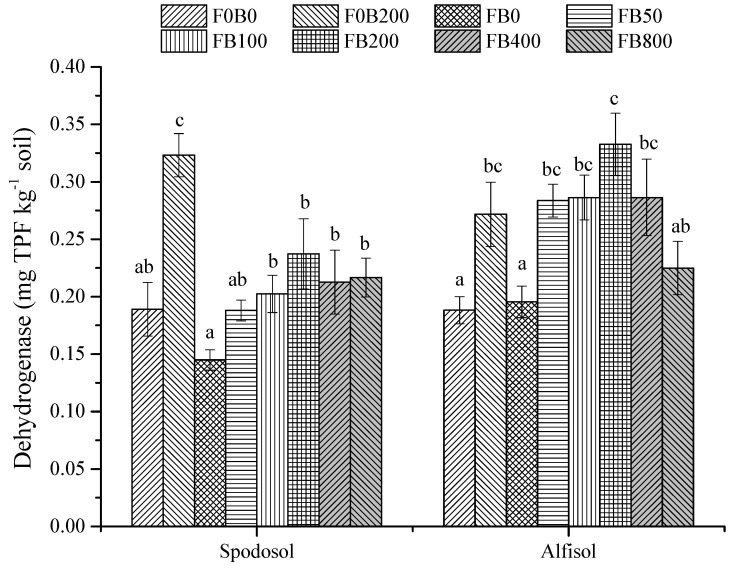
Soil dehydrogenase activity in Spodosol and Alfisol soils affected by different application rates of CNPs. F0B0, control (without fertilizers and CNPs); F0B200, without fertilizers and with CNPs at 200 mg kg^−1^ soil; and FB0, FB50, FB100, FB200, FB400 and FB800 represent those with and CNPs at application rates of 0, 50, 100, 200, 400 and 800 mg kg^−1^ soil, respectively. Error bars represent the standard error of the mean (*n* = 5). The letters a, b and c indicate statistically significant difference at *p* < 0.05 within each group comparison between the treatments. Bars with no common letters are significantly different (*p* < 0.05).

**Table 1 nanomaterials-11-02717-t001:** Nutrient use efficiency of corn plants affected by different application rates of CNPs in Spodosol and Alfisol soils.

Items	N Use Efficiency (%)		P Use Efficiency (%)		K Use Efficiency (%)	
	Spodosol	Alfisol	Spodosol	Alfisol	Spodosol	Alfisol
FB0	2.45 ± 0.56 ^a^	36.1 ± 3.24 ^a^	1.67 ± 0.37 ^a^	16.7 ± 2.12 ^ab^	3.83 ± 0.32 ^a^	30.9 ± 2.81 ^a^
FB50	25.3 ± 3.79 ^b^	39.6 ± 2.06 ^a^	7.04 ± 1.33 ^ab^	16.5 ± 1.20 ^a^	14.7 ± 2.43 ^b^	36.8 ± 1.50 ^ab^
FB100	35.1 ± 2.75 ^c^	43.7 ± 2.34 ^ab^	9.93 ± 2.15 ^bc^	16.9 ± 1.26 ^ab^	19.7 ± 1.60 ^c^	38.1 ± 4.03 ^ab^
FB200	48.8 ± 4.89 ^d^	50.5 ± 3.47 ^b^	11.8 ± 0.77 ^c^	19.0 ± 1.53 ^ab^	26.7 ± 1.97 ^d^	40.4 ± 2.38 ^b^
FB400	24.0 ± 1.79 ^b^	41.9 ± 3.53 ^ab^	10.7 ± 1.51 ^c^	16.3 ± 1.94 ^a^	15.5 ± 1.58 ^b^	43.5 ± 4.71 ^b^
FB800	2.72 ± 0.53 ^a^	44.7 ± 3.62 ^ab^	3.72 ± 0.64 ^a^	22.4 ± 2.81 ^b^	5.11 ± 0.51 ^a^	44.7 ± 2.38 ^b^

Note: F0B0, control (without fertilizers and CNPs); F0B200, without fertilizers and with CNPs at 200 mg kg^−1^ soil; and FB0, FB50, FB100, FB200, FB400 and FB800 represent those with and CNPs at application rates of 0, 50, 100, 200, 400 and 800 mg kg^−1^ soil, respectively. Error bars represent the standard error of the mean (*n* = 5). The letters a, b, c, and d indicate statistically significant difference at *p* < 0.05 within each column comparison between the treatments. Means with no common letters are significantly different (*p* < 0.05).

## Data Availability

Data is presented in the paper and also available on request.

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
