# Peer review of "Use of Carbon Nanoparticles to Improve Soil Fertility, Crop Growth and Nutrient Uptake by Corn (Zea mays L.)"

_nanomaterials, 2021, doi:10.3390/nano11102717_

Round 1

Reviewer 1 Report

This paper contains an extensive study of the use of carbon nanoparticles (CNP) as additives to soil to improve the growth of corn plants. I have read with pleasure this nice piece of work, where experiments have been nicely planned, performed and reported and the interesting results derived therefrom are very clearly discussed. In my opinion this paper is a very good contribution to the field and can be published in Nanomaterials “nearly” as it is. Limited additions and changes are indeed required, which should improve the information content and the legibility of the paper. These are listed hereafter.

In the Materials and Methods section of Page 2, the chemical analysis of CNP is provided, and, disregarding the very minor presence of nitrogen, seems to suggest an “average” composition near C2HO. Thus, only 50% of the atoms is C, and a significant portion of the CNP is made up by H and O. This should be clearly stated in the text, and the possibility that physiologically active molecular entities (ethers; heterocycles; sugars?) are present should be clearly assessed. This can easily be done by extracting soluble materials and performing NMR or HPLC analyses. Perhaps the excellent performance of CNP must be attributed to these “contaminants”. As a future work, let me suggest a pretreatment of CNP to eliminate such compounds, if present at all, to compare “washed and rinsed” CNP with the pristine ones.

Additionally, a more exhaustive solid-state characterization of CNP by IR, XRD and solid-state NMR is required. SEM imaging would also be helpful.

Very minor modifications:

Page 2, Line 47: “good conductivity”, whether thermal, electronic or ionic, is in this context, irrelevant, and should be omitted.

Page 5, Figure 2: The negative labels in the y axis for the Root Dry Biomasses should be removed, as mass is a positive value.

Formatting, at least in my version, needs a thorough revision, particularly by inserting blank lines between figure captions and main text. Also, le MDPI logo in page 11 should disappear.

Author Response

Dear Editor and Reviewers,

Thank you so much for your comments and revisions on this manuscript, entitled “Use of carbon nanoparticles to improve soil fertility, crop growth and nutrient uptake by corn (Zea mays L.)”.

All the comments were carefully considered and incorporated into the revised manuscript. We also carefully addressed grammatical errors. The point-by-point responses to all the comments are as follows.

 Response to the comments are as follows:

#Reviewer 1

Comments and Suggestions for Authors

This paper contains an extensive study of the use of carbon nanoparticles (CNP) as additives to soil to improve the growth of corn plants. I have read with pleasure this nice piece of work, where experiments have been nicely planned, performed and reported and the interesting results derived therefrom are very clearly discussed. In my opinion this paper is a very good contribution to the field and can be published in Nanomaterials “nearly” as it is. Limited additions and changes are indeed required, which should improve the information content and the legibility of the paper. These are listed hereafter.

In the Materials and Methods section of Page 2, the chemical analysis of CNP is provided, and, disregarding the very minor presence of nitrogen, seems to suggest an “average” composition near C2HO. Thus, only 50% of the atoms is C, and a significant portion of the CNP is made up by H and O. This should be clearly stated in the text, and the possibility that physiologically active molecular entities (ethers; heterocycles; sugars?) are present should be clearly assessed. This can easily be done by extracting soluble materials and performing NMR or HPLC analyses. Perhaps the excellent performance of CNP must be attributed to these “contaminants”. As a future work, let me suggest a pretreatment of CNP to eliminate such compounds, if present at all, to compare “washed and rinsed” CNP with the pristine ones.

Additionally, a more exhaustive solid-state characterization of CNP by IR, XRD and solid-state NMR is required. SEM imaging would also be helpful.

Re: Thanks for your suggestion. We will take full consideration of your suggestions in the future research.

Very minor modifications:

Page 2, Line 47: “good conductivity”, whether thermal, electronic or ionic, is in this context, irrelevant, and should be omitted.

Re: Deleted. Thanks.

Page 5, Figure 2: The negative labels in the y axis for the Root Dry Biomasses should be removed, as mass is a positive value.

Re: Revised as suggested. Thanks.

Formatting, at least in my version, needs a thorough revision, particularly by inserting blank lines between figure captions and main text. Also, le MDPI logo in page 11 should disappear.

Re: Thanks for pointing this out. We have revised the format throughout the manuscript.

Reviewer 2 Report

Zhenli He et al. have supplied carbon nanoparticles at different rates to corn (Zea mays L.) and have studied their impact on crop growth (pant height and shoot/root biomass), nutrient uptake, nutrient availability in soils and soil enzyme activity. Overall, the manuscript is well-organized, the experimental design is appropriate and the conclusions are mainly supported by the results. I recommend the publication of this manuscript in this journal after addressing the following concerns.

1) How do the authors characterize the nanomaterials used in the experiments? They give the features of the materials in section 2.1 but they do not give how the materials were obtained or characterized. This information should be added.

2) In the spodosol soil, the treatment FB0 and FB800 show similar results in the different parameter studied (Plant height, shoo and root biomass, nutrient uptake, etc…) How do the author explain these results?

Author Response

Dear Editor and Reviewers,

Thank you so much for your comments and revisions on this manuscript, entitled “Use of carbon nanoparticles to improve soil fertility, crop growth and nutrient uptake by corn (Zea mays L.)”.

All the comments were carefully considered and incorporated into the revised manuscript. We also carefully addressed grammatical errors. The point-by-point responses to all the comments are as follows.

Response to the comments are as follows:

#Reviewer 2

Comments and Suggestions for Authors

Zhenli He et al. have supplied carbon nanoparticles at different rates to corn (Zea mays L.) and have studied their impact on crop growth (pant height and shoot/root biomass), nutrient uptake, nutrient availability in soils and soil enzyme activity. Overall, the manuscript is well-organized, the experimental design is appropriate and the conclusions are mainly supported by the results. I recommend the publication of this manuscript in this journal after addressing the following concerns.

1) How do the authors characterize the nanomaterials used in the experiments? They give the features of the materials in section 2.1 but they do not give how the materials were obtained or characterized. This information should be added.

Re: Added as suggested. Please check lines 64-67 “The tested CNPs are composed of 62% C, 1.5% H, 35% O, 1.5% N, which was characterized by a VARIO EL III CHON analyzer (Elementar, Germany). The CNPs have a size range of 20-130 nm with negative charge on the surfaces (zeta potential -67.6 mV), which was determined using Malvern Nano-ZS (Malvern Instruments Ltd., Malvern, UK ).

2) In the spodosol soil, the treatment FB0 and FB800 show similar results in the different parameter studied (Plant height, shoo and root biomass, nutrient uptake, etc…) How do the author explain these results?

Re: According to present study, the slight growth inhibition only occurred at the highest application rate (FB800) in Spodosol soil, while FB800 significantly promoted plant growth in Alfisol soil The Spodosol soil has a lower buffering capacity than the Alfisol soil due to its low clay content, and therefore is more likely to present adverse dose-dependent effects of CNPs on crops. Further research is needed to understand this physiological response of crop plants to CNPs, especially at relatively high application rates.

Sincerely,

All coauthors

Round 2

Reviewer 2 Report

Authors address all the points of the last revision. Thus, I think that the present form of the manuscript is appropiate for publication.